

# Urban public bicycle dispatching optimization method

Fei Lin[1], Yang Yang[1,2], Shihua Wang[1], Yudi Xu[1], Hong Ma[1] and Ritai Yu[1]

[1] School of Computer Science and Technology, Hangzhou Dianzi University, Hangzhou, China
[2] Current affiliation: Institute of Intelligent Media Technology, Communication University of Zhejiang, Hangzhou, China

## ABSTRACT

Unreasonable public bicycle dispatching area division seriously affects the operational efficiency of the public bicycle system. To solve this problem, this paper innovatively proposes an improved community discovery algorithm based on multi-objective optimization (*CDoMO*). The data set is preprocessed into a lease/return relationship, thereby it calculated a similarity matrix, and the community discovery algorithm *Fast Unfolding* is executed on the matrix to obtain a scheduling scheme. For the results obtained by the algorithm, the workload indicators (scheduled distance, number of sites, and number of scheduling bicycles) should be adjusted to maximize the overall benefits, and the entire process is continuously optimized by a multi-objective optimization algorithm *NSGA2*. The experimental results show that compared with the clustering algorithm and the community discovery algorithm, the method can shorten the estimated scheduling distance by 20%–50%, and can effectively balance the scheduling workload of each area. The method can provide theoretical support for the public bicycle dispatching department, and improve the efficiency of public bicycle dispatching system.

## INTRODUCTION

With the progress of urbanization, people's awareness of low carbon life and health is increasing. The public bicycle system can provide a green and healthy way to travel, and gradually become an important part of the public transport system. However, the study of the division of public bicycle dispatching area is still in the primary stage. The division of the public bicycle scheduling area has two purposes: decomposing the scheduling between large-scale sites, and reducing the computational complexity of path planning.

At present, the mainstream regional division method is based on the urban administrative area, and each area is an independent scheduling area. However, the boundaries of residents' travel are not as clear as the administrative areas. With the development of the city, the links between the areas are more closely related, so the division based on urban administrative areas is lack of scientific basis. *Tulabandhula & Bodas (2018)* proposed a passenger monitoring system for dispatching vehicles in a public transportation network,

Corresponding author
Yang Yang, 162050103@hdu.edu.cn, yangyang_hdu@163.com

it monitors passengers at the station, vehicle scheduling information and processed hardware equipment. Because the size and population density of each administrative area are different, the number of sites in each area varies greatly. *Pan, Jun-Yi & Min (2015)* designed a heuristic simulated annealing hybrid search algorithm for large-scale VRP distributed problems. Firstly, based on the actual road network of GIS, the mathematical model is established. Secondly, the large-scale VRP path planning problem is studied. The administrative area is large in size and concentrated in population. *Forma, Raviv & Tzur (2015)* considers the spatial nature of public bicycle rentals, and the original inventory factor of bicycles. Then the paper establishes a regional maximum diameter distance constraint model. Finally, the best classification results are obtained by heuristic algorithms to minimize the overall inventory cost. Therefore, there are more sites, public bicycle turnover is high, and dispatching workload is large; but if there are fewer sites, public bicycle turnover is low, and dispatching workload is small. Above all, lack of a scientific planning method often leads to higher scheduling capital costs. *Schuijbroek, Hampshire & Hoeve (2016)* applied the maximum algebra algorithm to the division of the public bicycle scheduling area, and the paper established the corresponding partition mathematical model. The goal of zoning is to minimize the maximum completion time based on a reasonable level of service.

Aiming at the problems, this paper proposes an improved community discovery algorithm based on multi-objective optimization. By using this innovative algorithm, the results show that the algorithm brings three major benefits: it can effectively shorten the public bicycle scheduling distance, improve the scheduling efficiency, and effectively balance the workload of regional scheduling.

## RELATED WORK

The division of the public bicycle dispatching area involves operational research, and researchers have made significant contribution. Public bicycles and buses, as well as cargo transport vehicles are public transport, and their operations have similarities. Therefore, they can learn dispatching methods from each other. Kloimllner (*Miranda-Bront et al., 2017*) decomposes the problem of public bicycles into two sub-problems: scheduling area partitioning and scheduling path planning. Then create an integer programming model to achieve as few bicycle rental points as possible.

In addition, other researchers chose to use clustering algorithms. *Phanikrishnakishore & Madamsetti (2014)* used the rental rules between public bicycle stations, the space of public bicycle stations, and the non-spatial attributes of public bicycle stations, as well as the self-flow characteristics, using association rules to classify sites with strong correlation into the same category. Finally, various types of site space enclosed areas serve as the scheduling area for public bicycles. *Zhang, Liang & Wei (2017)* proposed a public bicycle scheduling area division scheme based on the improved *K-means* clustering algorithm. In the data analysis, the algorithm effectively estimates the $k$ central sites at the initial central site. After the *K-means* clustering algorithm is divided, the edge sites are clustered and

adjusted again according to the scheduling requirements. *Xu, Qin & Ma (2017)* integrates the spatial relationship between the sites, and the lease relationship of the bicycle, establishes the similarity matrix of the site, and proposes the parameters of the regional coupling, quantifies the degree of connection between the regions, and finally uses the clustering algorithm to obtain the corresponding result. *Long, Szeto & Huang (2014)* established a dynamic regional scheduling model, for large-scale public bicycle scheduling problems, and proposed a multi-stage re-optimized dynamic clustering algorithm, integrates optimal division, task balance between regions and regions. Within the balance of demand, three factors are progressively clustered, and in the process of solving, the abnormal sites are continuously split to gradually improve the clustering results. *Dziauddin, Powe & Alvanides (2015)* has studied the public bicycle dispatching area, found that there are often abnormal sites in the division, and he proposed a *K-Center* algorithm, adaptively limits the capacity of the rental site. *Hartmann Tolić, Martinović & Crnjac Milić (2018)* analyzed the spatial attributes and community structure of public bicycles, and the paper used the community discovery algorithm to analyze the community structure of public bicycles in Washington, London and Boston, and verified the existence of community structure in the public bicycle network.

The main method of scheduling area division is model method (*Dubey & Borkar, 2015*) and clustering algorithm (*Sun, Zhang & Du, 2015*). The model method requires abstract research, and there are many constraints and it is not easy to solve. Clustering algorithm is very difficult to determine the number of clusters, and it is difficult to evaluate. Moreover, the scheduling workload has no evaluation criteria, and it does not consider whether the workload is balanced. Therefore, this paper proposes a new method to solve the problem.

## SCHEDULING AREA DIVISION MODEL DESIGN

This part establishes the division model of public bicycle scheduling area, including the description of the model, and the assumptions of some conditions, and some interpretations of the parameters. Finally, this chapter will propose a lease/return point demand forecasting model, the data obtained from this model can help this paper verify whether **CDoMO**'s estimated total dispatch distance is the shortest.

### Problem description

At present, the clustering algorithm is mainly used to solve the problem of scheduling area division. The data set abbreviated to **DS** is preprocessed using a data preprocessing program. Turn a data set into a lease/return relationship abbreviated to **LRR** between sites. Then, through the similarity calculation between the sites, the similarity matrix abbreviated to **SM** is generated (*Yanping, Decai & Duoning, 2017*). Conversion from **DS** to **SM**, as shown in Eq. (3.1), where $R_{ij}$ represents the similarity between site $i$ and site $j$, $Q_{ij}$ represents the number of bicycles rented from the site $i$ and returned to the site $j$, $Q_{ji}$ represents the number of bicycles rented from the site $j$ and returned to the site $i$.

$$DS \overset{c_1}{\to} LRR = \begin{bmatrix} Q_{11} & Q_{12} & \cdots & Q_{1j} \\ Q_{21} & Q_{22} & \ddots & Q_{2j} \\ \vdots & \vdots & \ddots & \vdots \\ Q_{i1} & Q_{i2} & \cdots & Q_{ij} \\ Q_{11} & Q_{12} & \cdots & Q_{1i} \\ Q_{12} & Q_{22} & \ddots & Q_{2i} \\ \vdots & \vdots & \ddots & \vdots \\ Q_{j1} & Q_{j2} & \cdots & Q_{ji} \end{bmatrix} \overset{c_2}{\to} SM = \begin{bmatrix} R_{11} & R_{12} & \cdots & R_{1j} \\ R_{21} & R_{22} & \ddots & R_{2j} \\ \vdots & \vdots & \ddots & \vdots \\ R_{i1} & R_{i2} & \cdots & R_{ij} \end{bmatrix} \tag{3.1}$$

The conversion process represented by $c_1$ and $c_2$ is as follows: Eqs. (3.2), (3.3), $M$ represents the time range, which is based on the number of days:

$$c_1 : \text{progressing program} \tag{3.2}$$

$$c_2 : R_{ij} = \frac{Q_{ij} + Q_{ji}}{M} \tag{3.3}$$

$$SM = \begin{bmatrix} R_{11} & R_{12} & \cdots & R_{1j} \\ R_{21} & R_{22} & \ddots & R_{2j} \\ \vdots & \vdots & \ddots & \vdots \\ R_{i1} & R_{i2} & \cdots & R_{ij} \end{bmatrix} \overset{CA}{\to} DR = \{R_1, R_2, \ldots, R_n\} \tag{3.4}$$

Finally, the clustering algorithm abbreviated to $CA$ is used for dividing, $R_n$ stands for dividing into $n$ independent scheduling areas is shown in Eq. (3.4). If the division result abbreviated to $DR$ conforms to the lease/return law abbreviated to $LRL$, the user can actively complete a part of the scheduling work to reduce the scheduling workload. However, in the actual scheduling area division, in order to obtain the highest comprehensive benefits, the regional division should not only conform to the law, but also achieve the balance of scheduling workload as much as possible (*Shpak et al., 2017*). The regional scheduling workload is mainly determined by the distance within the area and the number of stations in the area. $Z_1$ and $Z_2$ should be as small as possible if the regional workload is balanced. This balance problem can be transformed into a multi-objective optimization problem. The objective function $f$ is shown in Eq. (3.5):

$$DR = \{R_1, R_2, \ldots, R_n\} \overset{MOO}{\to} \min f = [Z_1, Z_2]^T \tag{3.5}$$

$Z_1$ : Variance of the dispatch distance $Z_2$ : variance of the number of sites
    $MOO$: Multi-objective optimization
    Calculation of $Z_1$ in the following Eq. (3.6), $n$ represents the number of areas, $D_i$ represents the estimated dispatch distance of area $i$, and $\overline{D}$ represents the average of the estimated dispatch distances:

$$Z_1 = \frac{1}{n-1} \sum_{i=1}^{n} (D_i - \overline{D})^2 \tag{3.6}$$

Calculation of $Z_2$ in the following Eq. (3.7), $n$ represents the number of areas, $N_i$ represents the number of internal sites in area $i$, and $\overline{N}$ represents the average number of internal sites:

$$Z_2 = \frac{1}{n-1} \sum_{i=1}^{n} \left( N_i - \overline{N} \right)^2 \qquad (3.7)$$

$s.t.$

$$S = \left[ (S_i - P) \cup (S_j - P) \right] \cup P \qquad (3.8)$$

Equation (3.8) indicates that each site must be divided into an area. $S_i$ and $S_j$ represent the partition set. $P$ represents the parking lot sites collection:

$$\left[ [S_i - P] \cap [S_j - P] \right] = \varnothing (i \neq j) \qquad (3.9)$$

Equation (3.9) indicates that a site can only be divided into an area:

$$S_i \cap P \neq \varnothing, S_j \cap P \neq \varnothing \qquad (3.10)$$

Equation (3.10) indicates that each scheduling area contains at least one dispatch center. There are two optimization goals for this issue:

- **Minimize** the variance between the estimated dispatch distance between each area;
- **Minimize** the variance between the numbers of sites in each area.

## Model assumptions and parameter description

The scheduling area dividing process is complicated, and the abstract model involves many parameters. In order to make the model as close as possible to the actual division, before the model is established, some assumptions about the scheduling area dividing process are assumed:

- The scheduling distance of each area can be estimated theoretically, the estimated scheduling distance is approximately equal to the actual scheduling distance;
- Dispatching vehicles are not limited by driving time and mileage;
- Only one dispatching vehicle in each area is responsible for bicycle dispatch;
- Model of the dispatching vehicle is consistent with all parameters;
- The dispatching vehicle departs from the dispatching center, completes the dispatching task, and then returns to the original dispatching center, regardless of vehicle failure, and other unexpected factors.

Based on the problem description and model assumptions, the parameters and variables of the model in Table 1 are defined.

## Leasing demand forecasting model

After the scheduling area is divided, in order to calculate the estimated total distance of the scheduling, it is necessary to ensure that the demand for the lease/return site is known, so it is necessary to predict the scheduling demand for the lease/return site in the future. This section will be divided into **24-time** periods in hours per day named $t$, $t \in \{0, 1, \ldots, 23\}$. A Meteorology Similarity Weighted K-Nearest-Neighbour (**MSWK**) method is introduced to predict the number of least and returned bicycle at the site.

**Table 1  Parameters and variables of the model.** Based on the problem description and model assumptions, the parameters and variables of the model are defined.

| Parameters/variables | Parameter/variable meaning |
|---|---|
| $n$ | The number of areas |
| $i, j$ | Area number |
| $D_i$ | The estimated scheduling distance of the area $i$ |
| $D_j$ | The estimated scheduling distance of the area $j$ |
| $\overline{D}$ | Regional estimated dispatch distance average |
| $N_i$ | The number of sites in area $i$ |
| $N_j$ | The number of sites in area $j$ |
| $\overline{N}$ | Average number of sites within the area |
| $S$ | Collection of sites |
| $S_i$ | Site division set for area $i$ |
| $S_j$ | Site division set for area $j$ |
| $P$ | parking lot site collection |

**Table 2  Exact values.** In the measurement of the similarity of weather, the weather is split into five levels and assigned corresponding values. The exact values are shown in the table.

| Weather | Value |
|---|---|
| Heavy snow, heavy rain | 1 |
| Snow, light snow, moderate rain, light rain | 0.75 |
| Foggy | 0.5 |
| Sunny and cloudy | 0.25 |

### Leasing number forecast model

**MSWK** is an improved method for predicting lease/return bicycle quantity based on **KNN** algorithm. Analysed the amount of leasing in a similar time period to predict future leasing. Weather, temperature, humidity, winds speed, and visibility are measured in five indicators.

In the measurement of the similarity of weather, the weather is split into five levels and assigned corresponding values. The exact values are shown in Table 2.

The quantified weather conditions at $p$ and $q$ for two days $t$ is denoted by $W_{D_p^t}$ and $W_{D_q^t}$, respectively, and the weather similarities for $t$ in $p$ and $q$ are defined as follows (Eq. (3.11)):

$$\lambda_1 = \frac{1}{2\pi\sigma_1} e^{-\frac{\left(W_{D_p^t} - W_{D_q^t}\right)^2}{\sigma_1^2}} \tag{3.11}$$

The temperatures of the $p$ and $q$ two days $t$ periods are denoted by $F_{D_p^t}$ and $F_{D_q^t}$. The temperature similarities of the $t$ time periods in $p$ and $q$ are defined as follows (Eq. (3.12)):

$$\lambda_2 = \frac{1}{2\pi\sigma_2} e^{-\frac{\left(F_{D_p^t} - F_{D_q^t}\right)^2}{\sigma_2^2}} \tag{3.12}$$

The three dimensions of humidity, wind speed, and visibility are represented by a **3-D** Gaussian kernel function, and $H_{D_p^t}, S_{D_p^t}, V_{D_p^t}$ represents the humidity, wind speed, and visibility of the $t$ time period in $p$, respectively. The humidity, wind speed, and visibility similarity of $p$ and $q$ periods in $t$ are defined as follows (Eq. (3.13)):

$$\lambda_3 = \frac{1}{2\pi\sigma} e^{-\left(\frac{\left(H_{D_p^t} - H_{D_q^t}\right)^2}{\sigma_3^2} + \frac{\left(S_{D_p^t} - S_{D_q^t}\right)^2}{\sigma_4^2} + \frac{\left(V_{D_p^t} - V_{D_q^t}\right)^2}{\sigma_5^2}\right)} \tag{3.13}$$

In order to simplify the calculation, the temperature, humidity, wind speed, and visibility are normalized and all $\sigma_1, \sigma_2, \sigma_3, \sigma_4, \sigma_5$ are set to 1, thereby simplifying the calculation; finally, by weighting the above three similarity indexes, $p$ and $q$ can be obtained. The overall similarity indicator at time $t$ as follows (Eq. (3.14)):

$$M\left(D_p^t, D_q^t; a\right) = \delta_w\left(D_p^t, D_q^t\right) \sum_{i=1}^{3} a_i \lambda_i \tag{3.14}$$

Where $\delta_w\left(D_p^t, D_q^t\right)$ is a judgment function, when both $p$ and $q$ are working days or all non-working days, $\delta_w\left(D_p^t, D_q^t\right) = 1$, otherwise $\delta_w\left(D_p^t, D_q^t\right) = 0$. If you want to predict the amount of rent in the $t$ time period in $q$, select the most similar **K** days and use the **MSWK** algorithm to calculate the predicted value. The specific Eq. (3.15) is as follows:

$$s_i \cdot pd\left(D_q^t; a\right) = \frac{\sum_{p=1}^{K} M\left(D_p^t, D_q^t; a\right) s_i \cdot pd\left(D_p^t\right)}{\sum_{p=1}^{K} M\left(D_p^t, D_q^t; a\right)} \tag{3.15}$$

### Returning number forecast model

After a user rents a bicycle, they often return the bicycle to an adjacent site within a certain period of time. Therefore, there is a need for prediction data of the number of bicycles based on neighbouring sites, which is used to predict the number of bicycles returned to the site. Bicycles rented from site $i$ during time $t$ may be returned to site $j$ adjacent to $i$ during time $t$ or $t+1$. For the forecast of the number of return bicycles within the lease time $t$ period, it is necessary to first estimate the number of bicycles rented from the site $i$ and at the site $j$ within the time period $t$. The specific Eq. (3.16) is as follows:

$$e_{ij}^t = s_i \cdot pd(t) \frac{e_{ij} \cdot f}{s_i \cdot pd} \tag{3.16}$$

Among them, $s_i \cdot pd(t)$ is the predicted value of bicycle rental quantity from site $i$ in time period $t$, $e_{ij} \cdot f$ is historical record of bicycle rental from site $i$ and is still at site $j$. $s_i \cdot pd$ is historical total bicycle rental record from site $i$. Through the analysis of historical data, it is found that the user's riding time law can be fitted by the 2-Gaussian function. Therefore, the riding time $D_{ij}(t)$ between rental sites $i$ and $j$ can be estimated by Eq. (3.17):

$$D_{ij}(t) = g_1(t; \mu_1, \sigma_1) + g_2(t; \mu_2, \sigma_2). \tag{3.17}$$

Assume that the user's return time is evenly distributed, and the user's behaviour of returning the leased bicycle is completed within the $t$ time period or $t+1$-time period. During the time periods $t$ and $t+1$, the user $t_1$ rents a bicycle from the site $i$ at the moment, and the probability of returning the ticket at the site $j$ at $t_2$ is as follows (Eqs. (3.18) and (3.19)):

$$P_{ij}^{t} = \frac{1}{|t|} \int_{0}^{|t|} \int_{0}^{|t|-t_1'} dt_1' dt_2 D_{ij}(t_2) \tag{3.18}$$

$$P_{ij}^{t+1} = \frac{1}{|t|} \int_{0}^{|t|} \int_{|t|-t_1'}^{+\infty} dt_1' dt_2 D_{ij}(t_2). \tag{3.19}$$

Finally, considering the traffic patterns and the corresponding probabilities of the adjacent sites, the formula for predicting the number of return bicycles within the sites is obtained as follows:

$$s_i \cdot dd(t) = \sum_{j \neq i} e_{ij}^{t} P_{ij}^{t} + e_{ij}^{t-1} P_{ij}^{t+1}. \tag{3.20}$$

So far, the demand $\Delta N$ of the site $i$ at the time $t$ in the future will be calculated by combining the demand for rental and return of the rental site $i$ at the time $t$ in the future. The specific formula is as follows:

$$\Delta N = s_i \cdot dd(t) - s_i \cdot pd(t). \tag{3.21}$$

If $\Delta N$ is less than zero, it means that the site $i$ will not be able to meet the user's bicycle rental demand at the time $t$ in the future, and it is necessary to dispatch the bicycle through dispatch (*Feng, Zhu & Liu, 2018*). If $\Delta N$ is greater than the number of parking spots at the leased site, it means that the site $i$ at the time $t$ in the future cannot satisfy the user's demand for returning the car. It is necessary to reduce the number of bicycles by scheduling.

## COMMUNITY DISCOVERY ALGORITHM BASED ON MULTI-OBJECTIVE OPTIMIZATION

Community discovery algorithm based on multi-objective optimization, which combines quantitative indicators of regional scheduling workloads, community discovery algorithms (*Shivach, Nautiyal & Ram, 2018*), and multi-objective optimization algorithms (*Mori & Saito, 2016*). Firstly, the **Fast Unfolding** community discovery algorithm (*Sun et al., 2018*) is performed based on the similarity matrix of the site. Secondly, the workload adjusts the results of the community discovery algorithm. Throughout the process, the results are continuously optimized through a multi-objective optimization algorithm.

### CDoMO scheduling workload analysis

Scheduling workload is an indicator to measure the workload of a dispatch line. The scheduling itself involves many fields, so there is no uniform standard (*Kim, Jeong & Lee, 2017*). The generalized scheduling workload is mainly determined by the scheduling distance, the delivery volume and the number of service outlets. The three parameters

are weighted and integrated, and the workload of the dispatching line can be quantified. Suppose $W$ is the generalized scheduling workload, $D$ is the driving distance ($km$), $N$ is the number of outlets (*pieces*), $S$ is the delivery amount (*pieces*), and $\rho_1, \rho_2, \rho_3$ is the driving distance weight, the delivery amount weight, and the service outlet quantity weight as follows Eq. (4.1):

$$W = \rho_1 \cdot D + \rho_2 \cdot N + \rho_3 \cdot S \qquad (4.1)$$

This paper combines generalized scheduling workload with public bicycles, and then obtains a quantitative formula for regional scheduling workload, $W_i$ is the scheduling workload of area $i$, and $D_i$ is the scheduling distance of area $i$, which is calculated by the maximum generation star algorithm. $N_i$ is the number of stations in area $i$, $S_i$ is the number of stations in area $i$, and $\rho_1, \rho_2, \rho_3$ is the corresponding weight coefficient as follows Eq. (4.2):

$$W_i = \rho_1 \cdot D_i + \rho_2 \cdot N_i + \rho_3 \cdot S_i \qquad (4.2)$$

Since the regional scheduling is based on all stations in the entire area, and in the scheduling area division stage, the waiting scheduling sites and scheduling quantities of each area are unknown, so in this paper, the impact of $S_i$ on the scheduling workload is ignored, that is, let $\rho_3 = 0$. So, the Eq. (4.3) can be simplified to:

$$W_i = \rho_1 \cdot D_i + \rho_2 \cdot N_i. \qquad (4.3)$$

In the quantitative formula of scheduling workload, the weight coefficient cannot be determined manually, but when the scheduling workload balance is satisfied, the estimated scheduling distance variance in each area, and the variance of the number of stations in each area should be as small as possible, so the scheduling balance the problem can be turned into a multi-objective optimization problem. The objective function is $\min f = [Z_1, Z_2]^T$. **NSGA2** is the most popular multi-objective genetic algorithm. **NSGA2** first genetically manipulates the population $P$ to obtain the population $Q$; then the populations are combined and then combined with non-inferior sorting and crowding distance sorting, and then a new population is established. Repeat the above process, until the termination condition is met. The detailed process is as follows:

(1) Randomly generate the initial population $P_0$, then sort the populations non-inferiorly, and assign a non-dominant value to each individual; then perform the operations of selection, crossover, and mutation on the initial population $P_0$ to obtain a new population $Q_0$, set to $i = 0$.

(2) Combine the populations of the father and offspring, then form a new population $R_i = P_i \cup Q_i$, and then sort the population $R_i$ non-inferiorly to obtain the non-inferior layer *F1, F2, ⋯*.

(3) Perform replication, crossover, and mutation operators on population $P_{i+1}$ to form population $Q_{i+1}$.

(4) If the termination condition holds, then it ends; otherwise, $i = i+1$, go to step (2). The main process diagram of **NSGA2** is shown in Fig. 1:

This paper uses the **NSGA2** multi-objective optimization algorithm to resolve the scheduling area partition model (*Wu, 2014*). The length of the chromosome in **NSGA2**

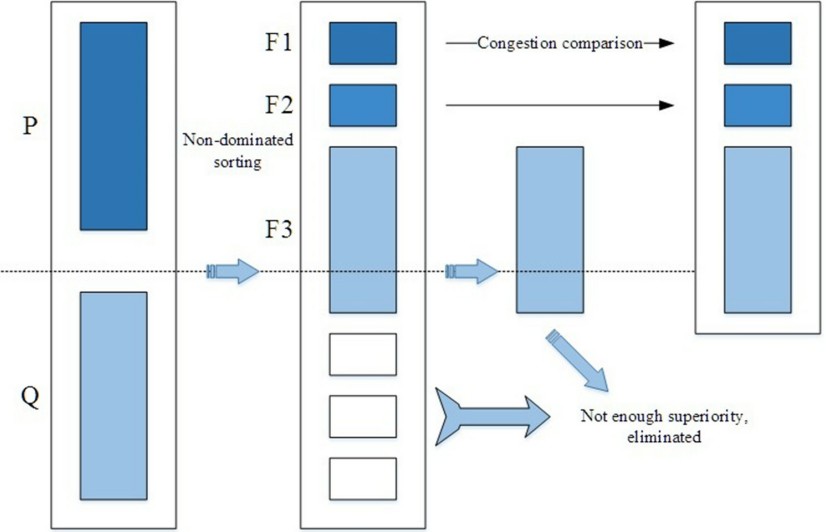

**Figure 1** **The main process of NSGA2 algorithm.**

is 2 (*Basch et al., 2015*), which corresponds to the value of the weight parameter in the area scheduling workload. Each individual corresponds to a scheduling workload formula, based on schedule the workload adjustment community found the results of the division. Figure 2 shows the restricted flow of *NSGA2* algorithm.

### CDoMO algorithm design

Community discovery algorithm built on multi-objective integrates quantitative indicators of regional scheduling workloads, community discovery algorithms and multi-objective optimization algorithms. Firstly, the *Fast Unfolding* community discovery algorithm is implemented based on the similarity matrix of the least sites; secondly, the workload index is used to adjust the results of the community discovery algorithm. The entire process continuously optimizes the results from a multi-objective optimization algorithm.

Table 3 displays the detailed algorithm calculation.

## EXPERIMENT AND ANALYSIS

The rest of the paper is part of the experiment and analysis. The experimental section was divided into two groups, which were experiments using New York public bicycle data and Chicago public bicycle data. In the analysis section, the two groups of experiments use *K-means* clustering algorithm, and *Fast Unfolding* community discovery algorithm as comparisons, it compares the three aspects of the number of rental sites, the variance of the number of scheduled bicycles, and the estimated total distance of scheduling. The comparative data show that the algorithm is effective against both sets of experiments.

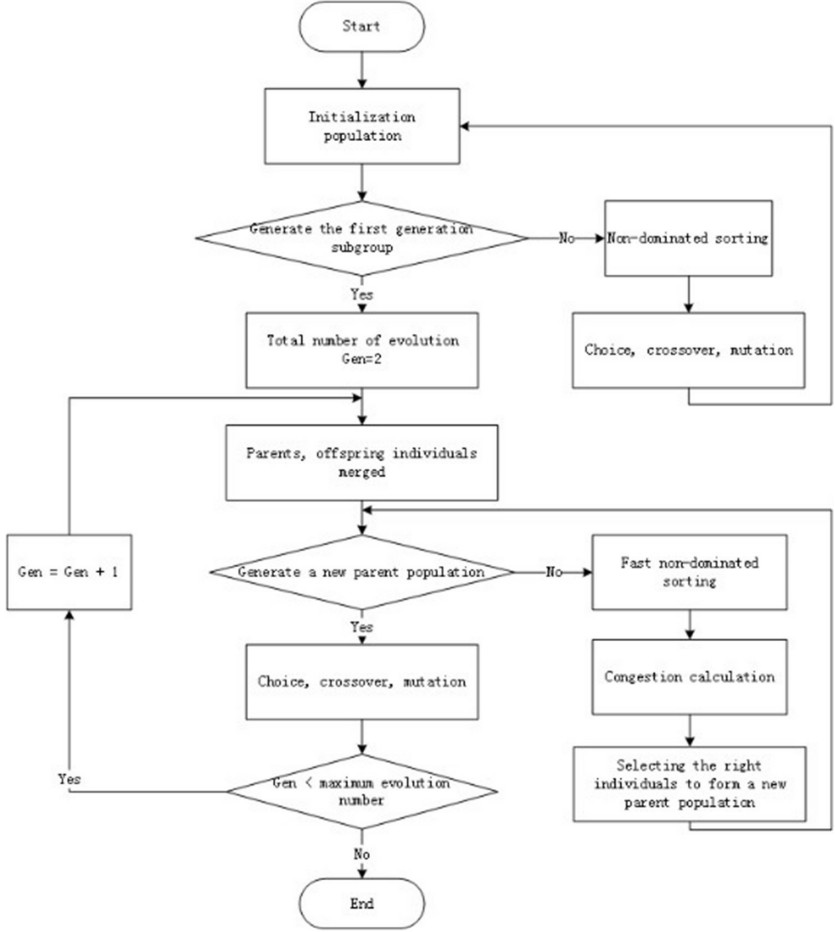

**Figure 2   Specific flow of NSGA2 algorithm.**

## New York public bicycle
### Data set introduction

Citi Bikes (*Jiang et al., 2018*) is a people-benefit project launched by the New York City Government. Figure 3 displays the spatial distribution of rental sites. Blue represents Manhattan, with 250 rental sites; green represents Brooklyn, with 77 sites. Each Citi Bicycle rental site has GPS location information, so it is not difficult to locate the rental site. The system records the user's data onto each cycle. The package contains the location and time data onto the start and the end of the site, the entire riding process, the bicycle ID, and the user's gender and birth date. This experiment will use the May 2016 rent-return dataset of New York public bicycles to conduct an experiment, a total of 96, and 1986 rent-return data. The dataset contains 16 fields, and the nine fields related to this experiment are shown in Table 4.

This paper uses the pre-processing program to process the leased data, it turned into the lease-return relationship between the least sites (*Guo et al., 2017*). It also generates a similarity matrix based on the rent-return relationship. The similarity calculation formula

**Table 3** Detailed algorithm calculation.

| Algorithm: | Community discovery algorithm based on multi-objective optimization |
|---|---|
| Input: | Site similarity matrix $X$, population number *popsize*, maximum number of iterations *MaxGen*. |
| Output: | Optimal regional division results $\rho_1^*$, and workload index parameters $\rho_2^*$. |
| 1. | Initialize the historical optimal solution $f^*$ and its workload index parameter $\rho_1^*, \rho_2^*$. |
| 2. | Perform a pass phrase of the Fast Unfolding community discovery algorithm, and obtain the results of the preliminary zoning division as $R$. |
| 3. | Calculate the estimated distance $D_i$ of each area in $R$, number of regional sites $N_i$. |
| 4. | Individual genes in the population as weight coefficients $\rho_1, \rho_2$. Finally, the scheduling workload of each area is calculated by the formula $W_i = \rho_1 \cdot D_i + \rho_2 \cdot N_i$. The variance of the regional workload is denoted as $V$. |
| 5. | For each rental site $i$, try to put $i$ into other communities and calculate the incremental $\Delta V$ of the adjustment workload, the entire process records the maximum $\Delta V_{max}$ and the corresponding community $k$. If $\Delta V_{max} < 0$, site node $i$ does not adjust; if $\Delta V_{max} > 0$, node $i$ is adjusted to community $k$. Traverse all the site until all the site are adjusted and the result is recorded as $R^*$. |
| 6. | Define the variance function $f_1$ of the regional site, and define the regional dispatch distance variance function $f_2$, they are two objective functions to perform fast non-dominated sorting on the results, the records of the optimal solution in the contemporary population as $f'$, and its corresponding scheduling workload parameters are denoted as $\rho_1', \rho_2'$. If $f' > f^*$ after comparison, letting $\rho_1^* = \rho_1', \rho_2^* = \rho_2'$. |
| 7. | Determine whether the number of program iterations exceeds the maximum number of iterations *MaxGen*. If it exceeds, the optimal regional division results, and workload index parameters $\rho_1^*, \rho_2^*$ are output; otherwise, a new population is generated through elite strategy selection, which can ensure that certain elite individuals will not be discarded during the evolution process, thereby improving the accuracy of the optimization results, and expanding the sampling space. And gene crossover and mutation processes and the execution continue from 1. |

for the least sites is as follows (Eq. (4.4)):

$$R_{ij} = \frac{Q_{ij} + Q_{ji}}{M} \tag{4.4}$$

Among them, $R_{ij}$ represents the similarity between site $i$ and site $j$; $Q_{ij}$ represents the number of times to rent a bicycle from site $i$ and site $j$ to return the bicycle; $Q_{ji}$ represents the number of times of renting a bicycle from site $i$ and returning it at site $j$; $M$ represents the time range in days. In this experiment, the data set was a total of 31 days in May 2016, so $M = 31$. The corresponding abstract network can be generated through the lease-return relationship (Fig. 4). Due to the dense population, dense sites, and prosperous business, the sites in Manhattan are more closely linked, and Brooklyn is a river is separated from Manhattan, so the connection between the two regional sites is sparse except for the leases along the river.

### Experimental result

In the experiment, we first used the Gephi visualization network analysis platform to analyse the community structure in the data (*Hu, An & Wang, 2018*). The Gephi platform uses the integrated Fast Unfolding algorithm, it divides the public bicycle abstraction network according to the rules of public bicycle rental. The Fast Unfolding algorithm mainly

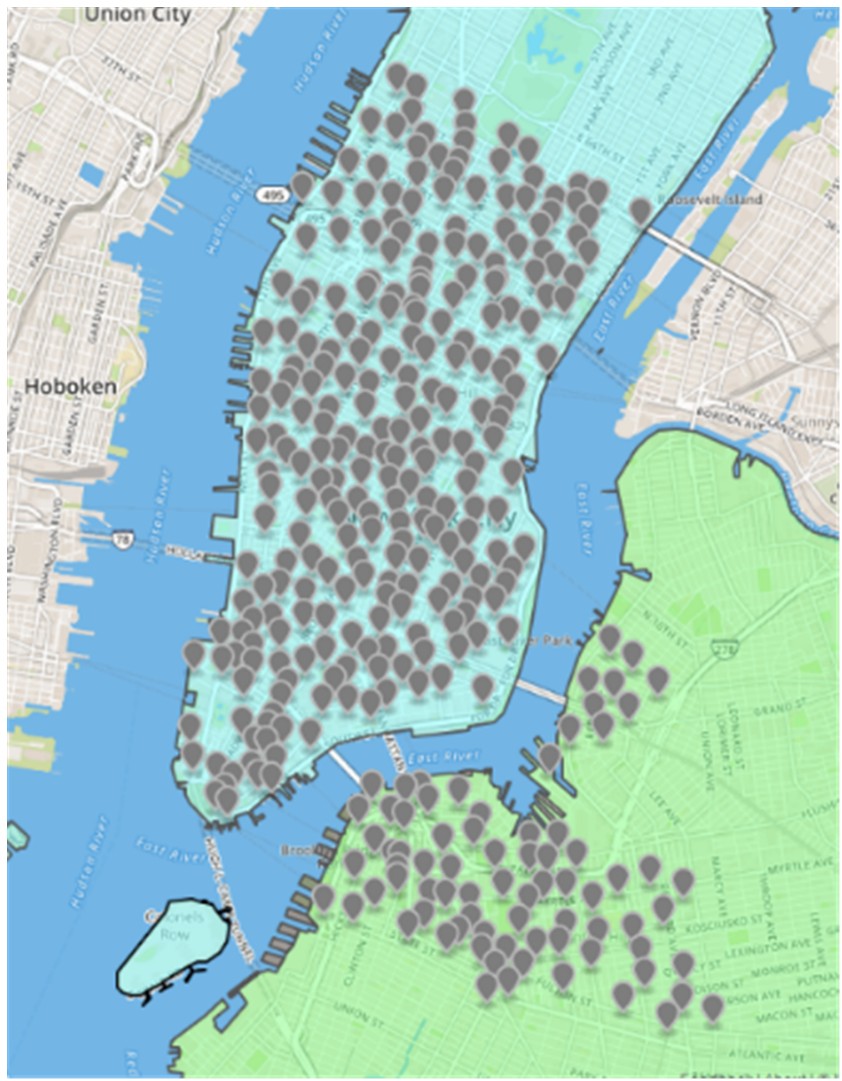

**Figure 3** Spatial distribution map of public bicycles in New York.

includes two phases. The first phase is known as Modularity Optimization. The main part is to divide each node into the community, its neighbourhood nodes are located, so that the value of the module degree becomes larger; the second phase is called community. Aggregation is mainly to aggregate the communities divided in the first step into one site, that is, to rebuild the network based on the community structure generated in the previous step. Repeat the above process, until the structure of the network no longer changes (Fig. 5).

After the *Fast Unfolding* algorithm for the New York public bicycle rental site in this paper, Fig. 6 shows the internal community structure of the abstract network of New York's public bicycles, where the dots represent sites, where the sites of different communities are represented by different colours, and the lines represent the relationships between the sites; obviously, six communities have more close contact with leases within the same community, and the links between different societies are relatively sparse. The results of the

**Table 4  Dataset contains 16 fields, and the nine fields.**

| No. | Fields | Meaning |
|---|---|---|
| 1 | start time | Starting time |
| 2 | stop time | End Time |
| 3 | start_station_id | Bicycle rental site ID |
| 4 | start_station_name | Name of bicycle rental site |
| 4 | start_station_longitude | Longitude of rental bicycle rental site |
| 5 | start_station_latitude | Latitude of rental bicycle rental |
| 6 | end_station_id | Return bicycle rental ID |
| 7 | end_station_name | The name of the bicycle rental site |
| 8 | end_station_longitude | The longitude of the bicycle rental site |
| 9 | end_station_latitude | The longitude of the bicycle rental site |

*Fast Unfolding* community discovery algorithm are mapped to map on New York (Fig. 7). Manhattan is a densely populated administrative district, and the vast majority of public bicycles in the area ride on the inside, so the Manhattan District is divided into five areas according to the law of rent. Brooklyn is structured in a district. Although the division results are relatively reasonable, there are still many abnormal sites. These abnormal sites are far away from their respective areas; the number of sites of each area is uniform.

*CDoMO* is based on community discovery algorithm, considering the regional scheduling workload factors. The regional scheduling workload is determined by estimated dispatch distance and the number of regional least sites. If the community finds out that there are abnormal sites, it will cause regional forecasting scheduling distance become larger, so that the variance between the scheduling distances will become larger. If there is a major difference in the number of sites between areas, the variance between the numbers of sites will increase. The goal of *CDoMO* is to optimize the variance of the distance between the regional scheduling, and optimize the variance of the number of sites. In the optimization process, the division results can be adjusted to make it more reasonable. The division process does not take into consideration the deficiencies in the workload balance in each scheduling area. After the community discovery algorithm based on multi-objective optimization solves the division model of the public bicycle scheduling area, the experimental results shown in Fig. 8 are obtained. By comparing the result shows that the sites along the Williamsburg Bridge and the riverside along Manhattan is divided into the same dispatch area, which is more n line with the rules of public bicycle rental and resolving the anomaly (*Zhang et al., 2011*). The difference between the number of sites and regional sites is too large (*Manju & Fred, 2018*).

In order to maintain the consistency of the experiment, the value of *k* in the classical clustering algorithm *K-means* algorithm is set to 6 (*Lin & Song, 2017*), and then the clustering is based on the same data set; the space area enclosed by the sites in each class as the scheduling area. In order to achieve regional division, the results of the regional division based on the clustering algorithm (Fig. 9). It was found that when the clustering number is *k*=6, the clustering algorithm achieves a poor regional division. The number of sites in the class represented by the red is very large, while the number of classes represented by purple

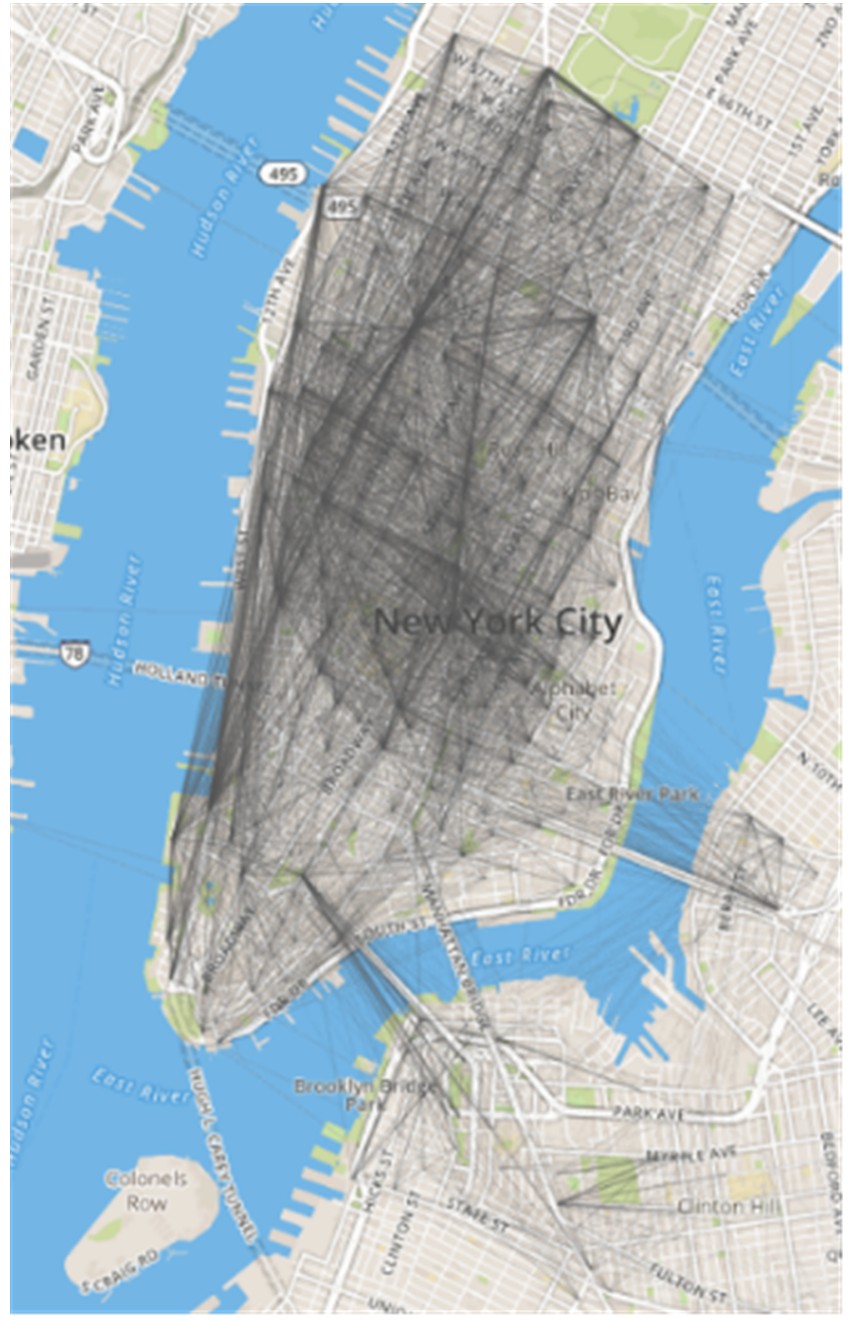

**Figure 4  Abstract network of public bicycles in New York City.**

and beige is very small, and the number of sites to vary greatly from the types. In addition, the boundaries of each scheduling area are unclear and are overlapped (*Zhen et al., 2016*).

### Algorithm performance comparison results

Built on the overall experimental results of the above three methods, it is found that the multi-objective optimization-based community discovery algorithm proposed to this

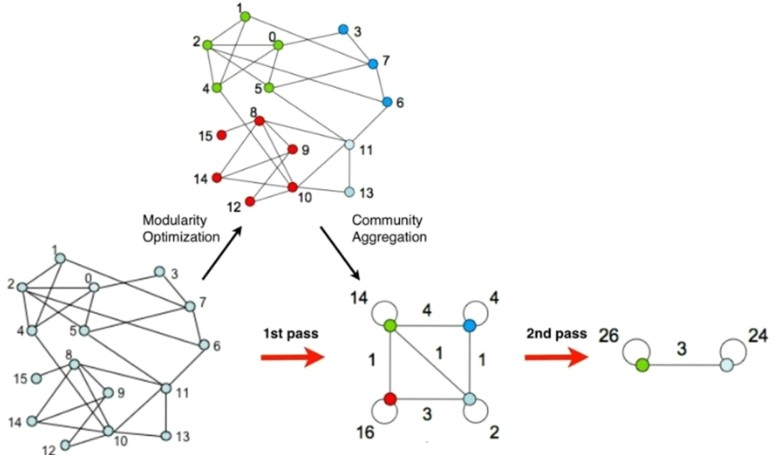

**Figure 5** Schematic diagram of community discovery algorithm.

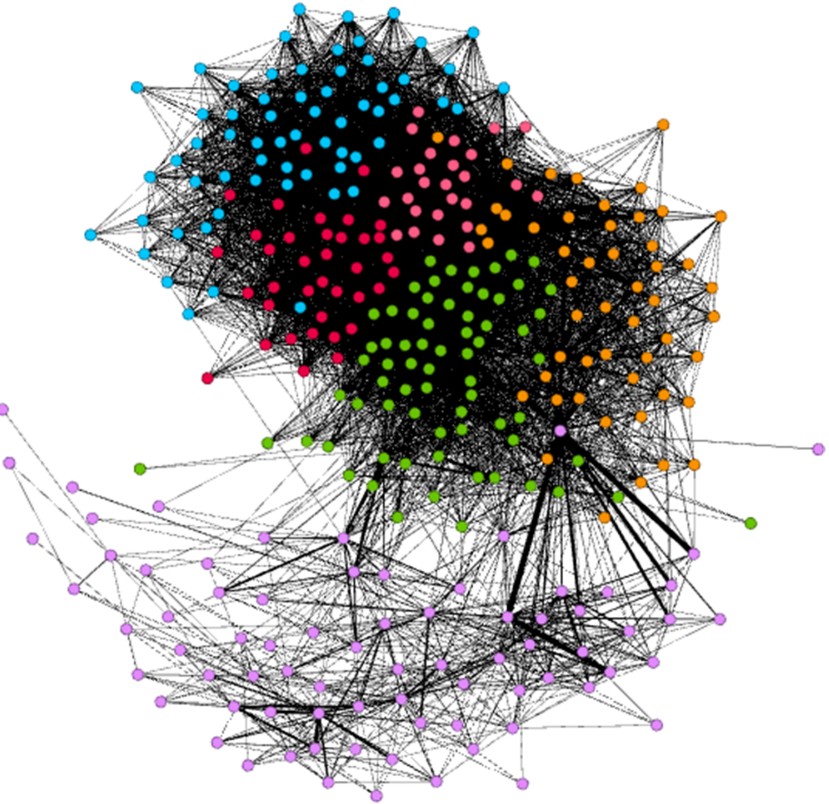

**Figure 6** Schematic diagram of analysis results of the Gephi Visual Network Analysis platform.

paper can make the division of the areas consistent with the rules and make the regional scheduling workload as balanced as possible. In addition to the analysis of the overall distribution of provincial division space, the paper also compares and analyses the three

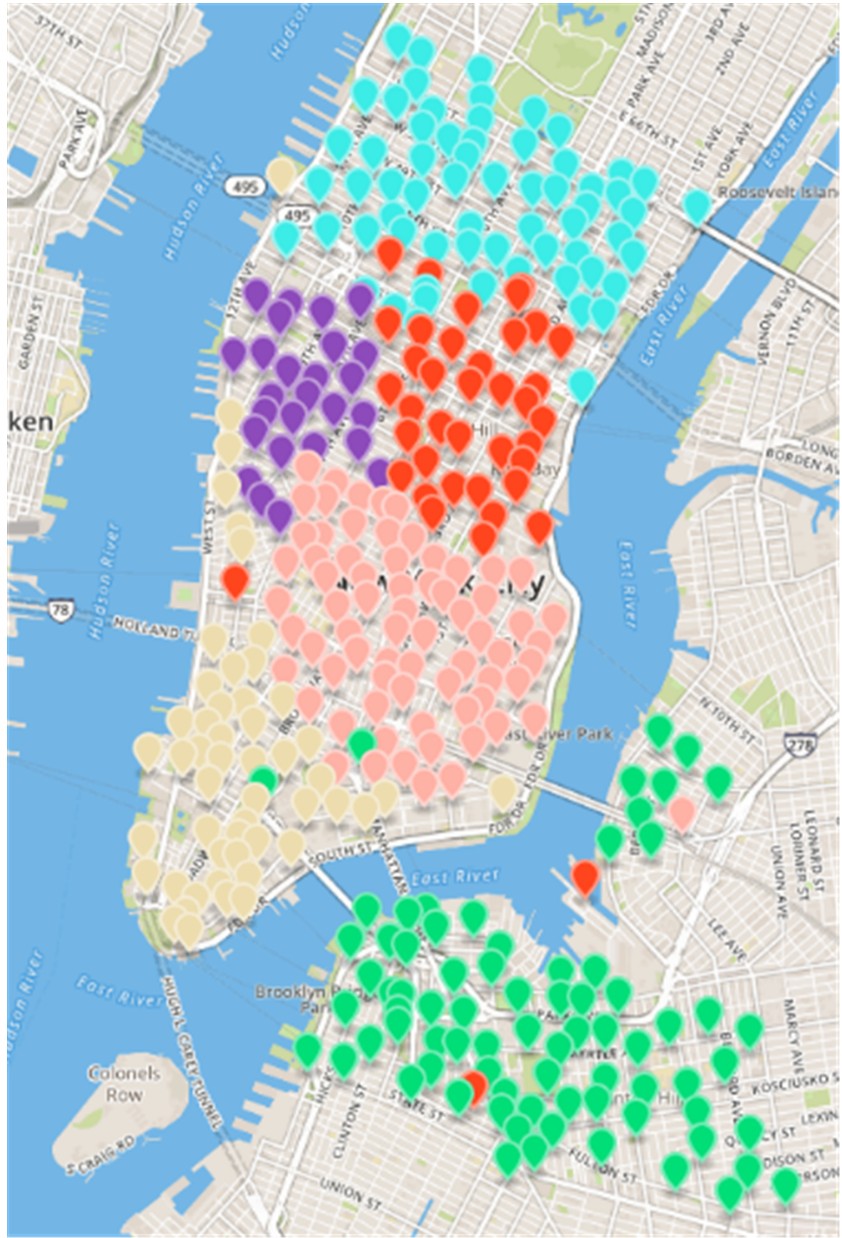

**Figure 7 Results of *fast unfolding* community Discovery algorithm in the New York dataset.**

dimensions of the regional rental site variance, the regional dispatch distance variance, and the estimated total dispatch distance. Figure 10 compares the variance between the numbers of sites. The data show that the variance between the ***CDoMO*** compared to the ***K-means*** algorithm is reduced by 63.31%, and the variance of the ***Fast Unfolding*** algorithm is reduced by 32.32%. Figure 11 compares the variance of the number of bicycles dispatched in the area. The data show that the variance of the ***CDoMO*** algorithm compared to the ***K-means*** algorithm is reduced by 88.06%, and the variance of the ***Fast Unfolding*** algorithm

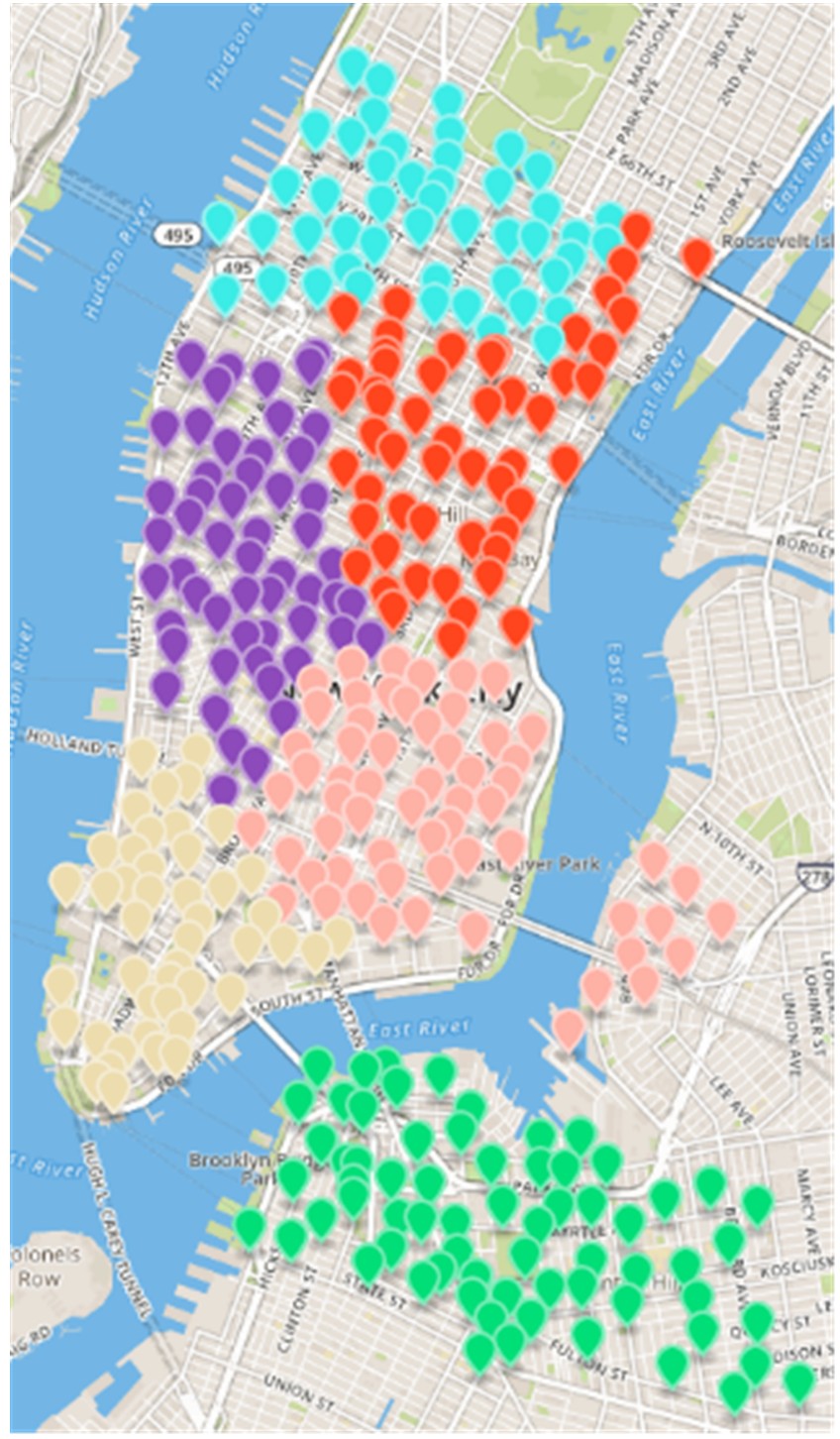

**Figure 8    Results of region partition based on multi-objective optimization algorithm in the New York dataset.**

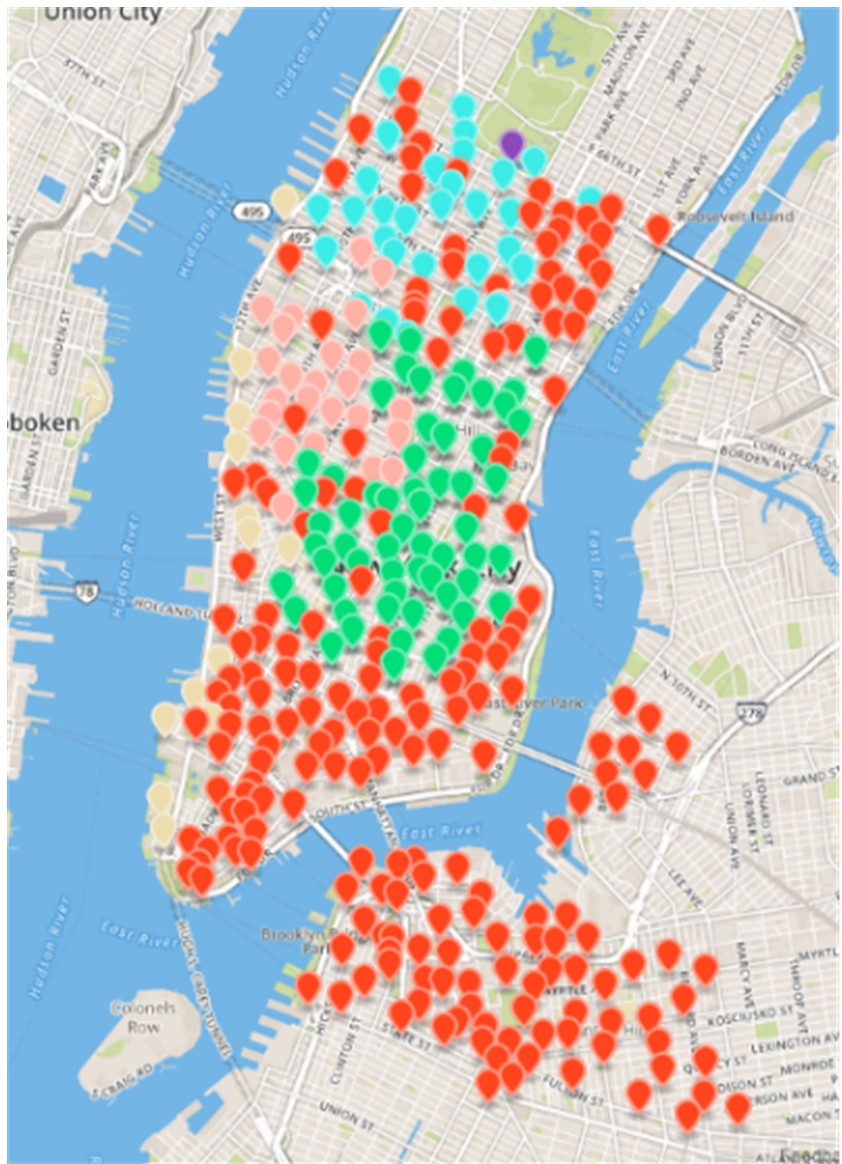

**Figure 9** **Results of region partition based on clustering algorithm in the New York dataset.**

is reduced by 38.14%. Figure 12 compares the estimated total scheduled distances. The data show that the variance of the *CDoMO* algorithm is 55.17% compared with the *K-means* algorithm and 27.54% compared to the *Fast Unfolding* algorithm. When scheduling and partitioning based on multi-objective optimization algorithm, the estimated scheduling distance can be shortened, and the estimated scheduling distance is positively related to the actual scheduling distance, so the actual scheduling distance will also be shortened; in addition, the scheduling work of each area will also be made. Relatively balanced. Figure 13 is a comparative display of experimental results.

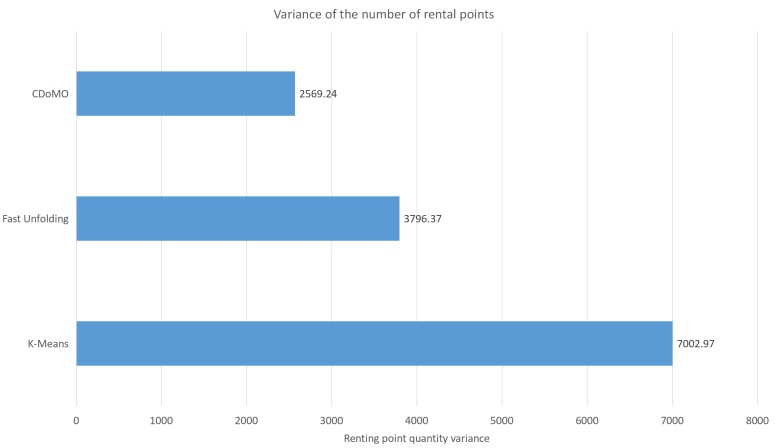

**Figure 10** **Difference in the number of bicycle rental points under three algorithms in the New York dataset.**

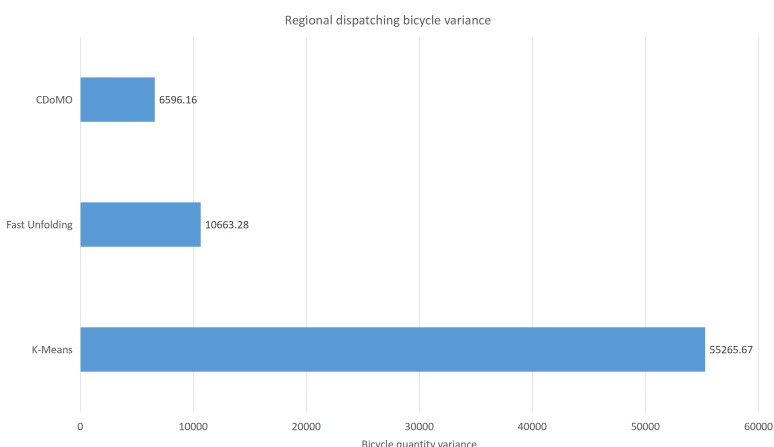

**Figure 11** **Distance difference of bicycle scheduling area under three algorithms in the New York dataset.**

## Chicago public bicycle
### Data set introduction

First of all, the data set cited in this paper is from Chicago public bicycle data (*Ye, Chu & Xu, 2015*). The starting site is 2015-1-1, and the deadline is 2015-6-30. There are two quarters and six months of data, a total of 759,789 data records. This paper did some data pre-processing: Trips that did not include a start or end date were removed from the original table. Then, in order to ensure that the information of the data set more abundant, this paper decided to use the data set, distance information of each pair of source address and destination address. Finally, we utilize certain data pre-processing methods to remove weather and other data because it can be considered as an ideal condition. The dataset

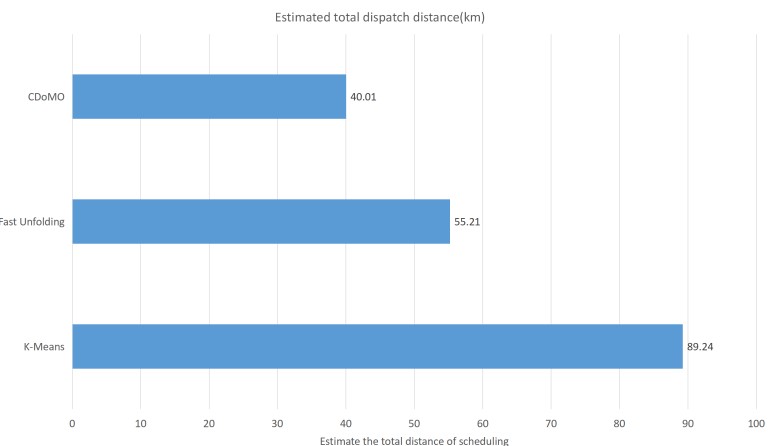

**Figure 12    Total distance of estimated scheduling under three algorithms in the New York dataset.**

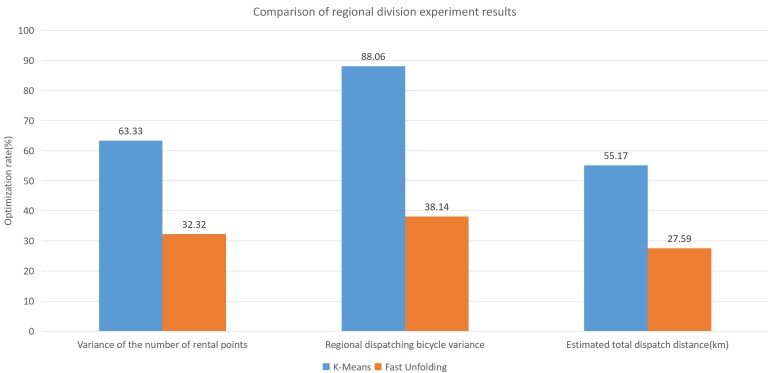

**Figure 13    Comparison of experimental results of region partition under three algorithms in New York dataset.**

contains 12 fields, and the 10 fields related to this experiment are presented in the following Table 5.

### Experimental result

Results of the *Fast Unfolding* community discovery algorithm can be mapped to Chicago map as the picture shows (*Ye, Chu & Xu, 2015*). In contrast, the division results are more uniform and reasonable, but there are too many abnormal sites in the middle. These abnormal sites are a long way from where they should have existed. The number of rental sites in a divided area is not particularly uniform (Fig. 14).

Based on *CDoMO*, in the optimization process, the division results are dynamically adjusted in time. Therefore, in this case, the division result is more reasonable, and the problem of scheduling balance, this algorithm obviously adds more consideration. It not only addresses the problem of abnormal sites, but also solves the problem of differences in the number of regional sites at the same time (Fig. 15). In order to make the experiment

**Table 5 Dataset contains 12 fields, and the 10 fields.**

| No. | Fields | Meaning |
| --- | --- | --- |
| 1 | start time | day and time trip started, in CST |
| 2 | stop time | day and time trip ended, in CST |
| 3 | from_station_id | ID of station where trip originated |
| 4 | from_station_name | name of station where trip originated |
| 5 | from_station_longitude | Longitude of rental bicycle rental site |
| 6 | from_station_latitude | Latitude of rental bicycle rental |
| 7 | to_station_id | ID of station where trip terminated |
| 8 | to_station_name | name of station where trip terminated |
| 9 | to_station_longitude | The longitude of the bicycle rental site |
| 10 | to_station_latitude | The longitude of the bicycle rental site |

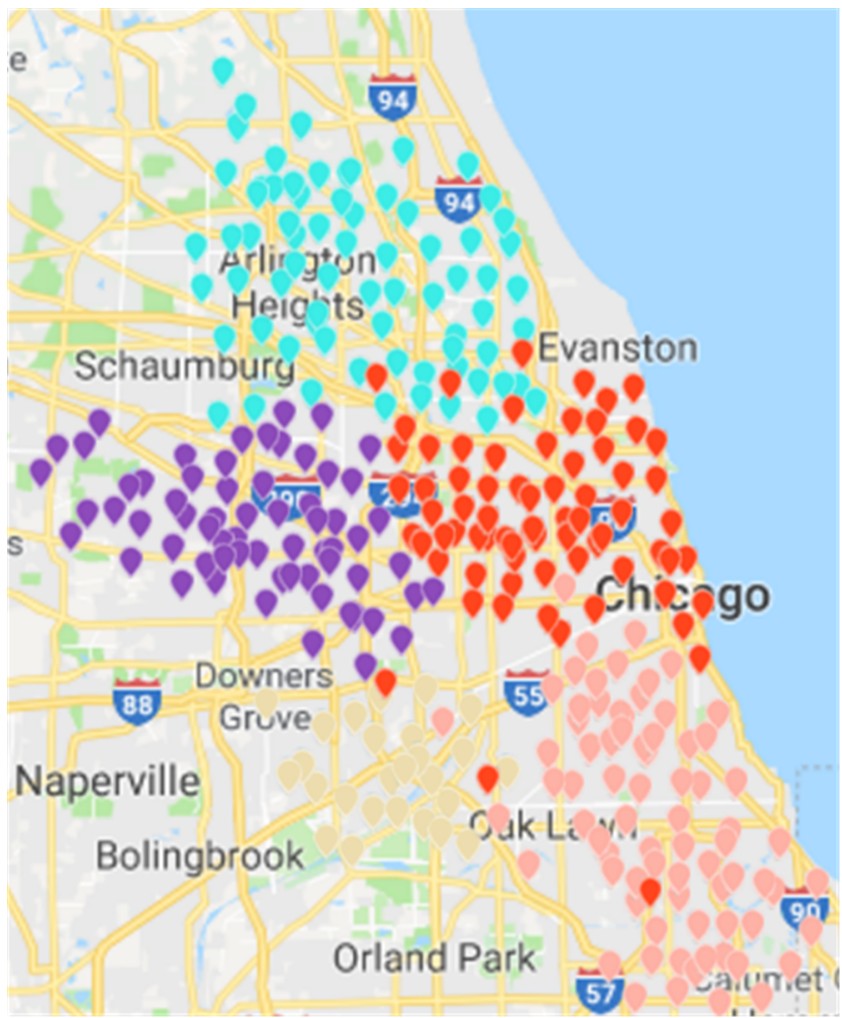

**Figure 14 Results of *fast unfolding* community discovery algorithm in Chicago dataset.**

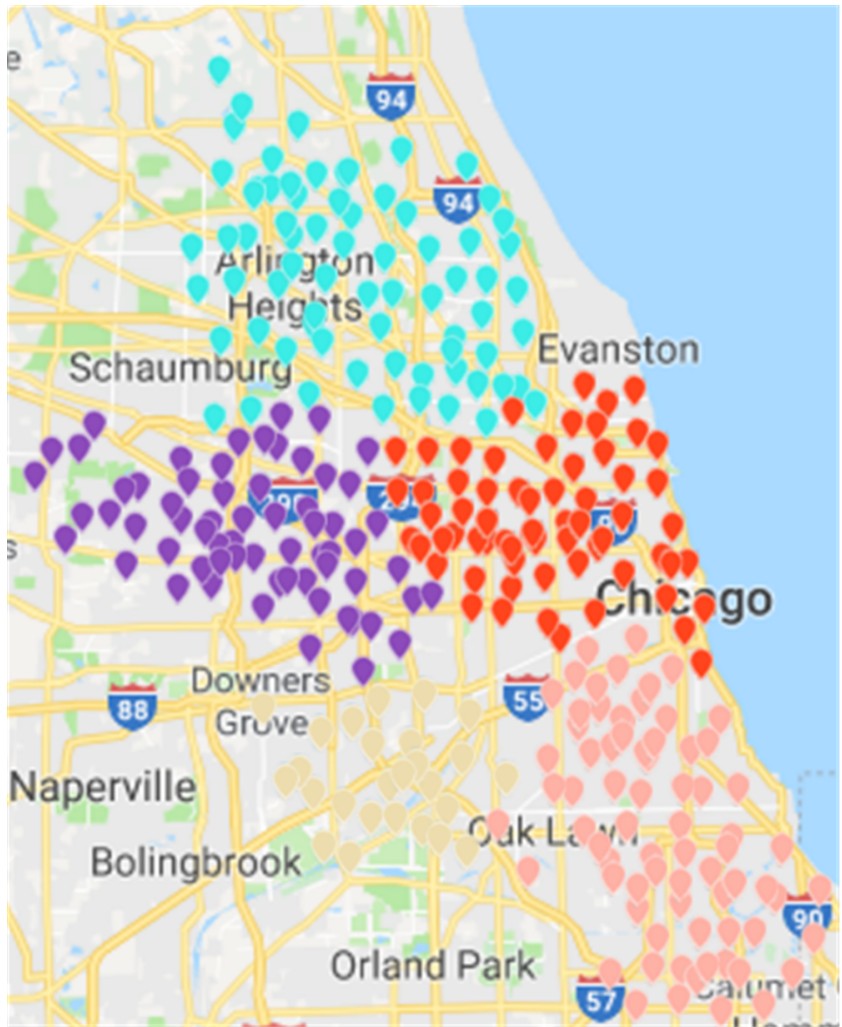

**Figure 15** **Results of region partition based on multi-objective optimization algorithm in the Chicago dataset.**

consistent, we set the value of **$k$** in the **K-means** algorithm as **5**, and then we clustered the uniform data set. The results of the clustering are presented in the figure. This paper believes that the results obtained by the clustering algorithm are very poor, because the red sites represent a particularly large number of sites. The yellow site represents a particularly small number of rental sites. This shows that the various types of leases, the number of differences is too large, in addition, this algorithm also led to the border is not clear, and there is some inevitable overlap. In the actual scheduling work, this situation is not allowed, as showed in Fig. 16.

### Algorithm performance comparison results

This paper will describe the quantified experimental results of the three methods, it compares the differences between them. It is easy to see that the algorithm proposed in

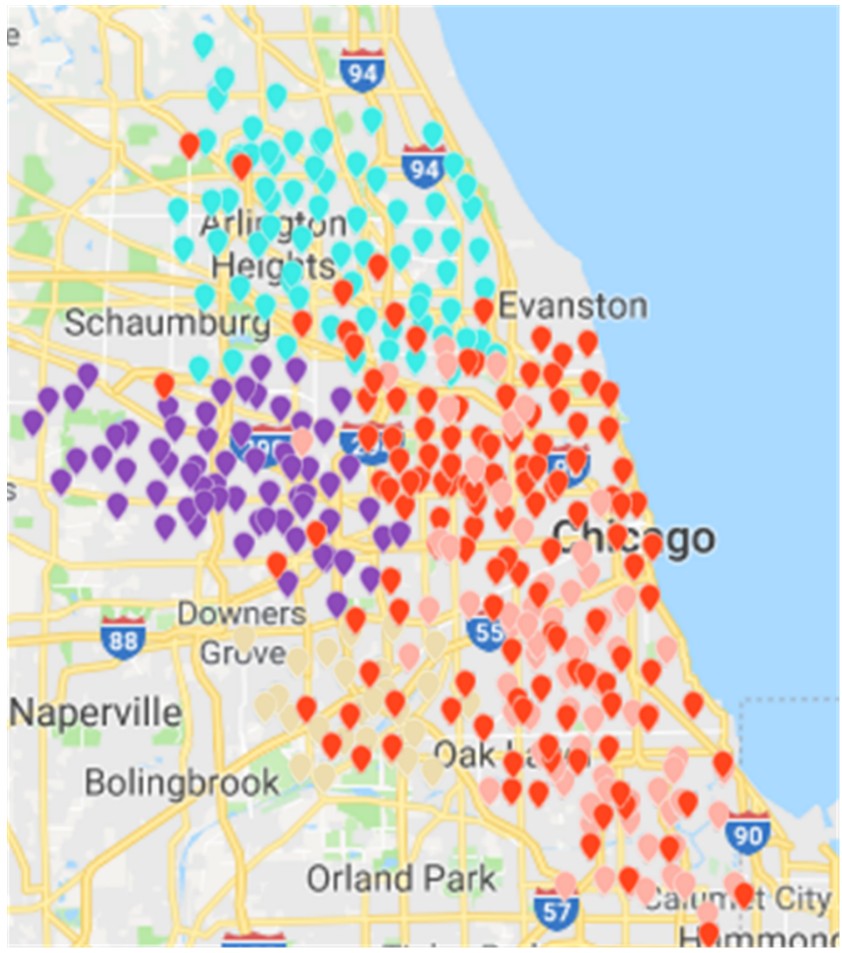

**Figure 16** Results of region partition based on clustering algorithm in the Chicago dataset.

this paper is optimal, compared to the other two algorithms. Figure 17 compares the difference between the numbers of sites. Compared to the ***K-means*** clustering algorithm and the ***Fast Unfolding*** community discovery algorithm, the variance of the ***CDoMO*** algorithm is reduced by 66.98% and 22.57%. Figure 18 in this paper compares the number of scheduled bicycles, it finds that the ***CDoMO*** algorithm set out in the present paper is an optimal algorithm. Similarly, opposed to the ***K-means*** clustering algorithm and the ***Fast Unfolding*** community finding algorithm, the variance is reduced by 83.77% and 48.72% (Fig. 19). Figure 20 compares the estimated total distance of scheduling with the other two algorithms, and the conclusion shows that the distance is decreased by 50.82% and 22.08%.

Then we can conclude that the ***CDoMO*** algorithm proposed in this paper: It effectively reduces the number of sites; it effectively reduced the variance in the number of bicycles dispatched; it effectively reduced the estimated total distance for scheduling.

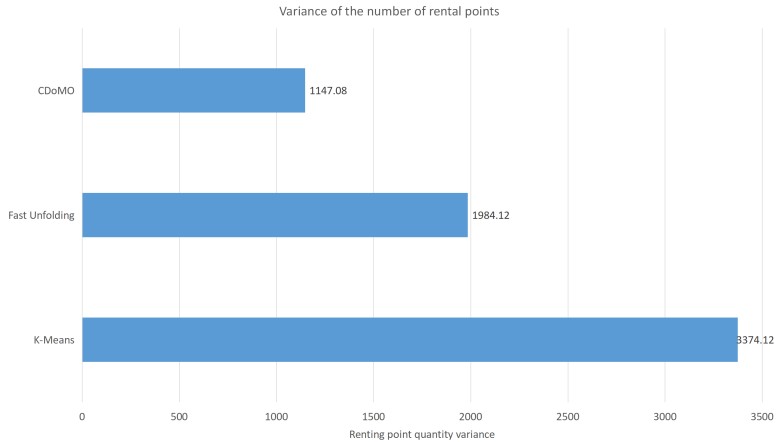

**Figure 17** Difference in the number of bicycle rental points under three algorithms in the Chicago dataset.

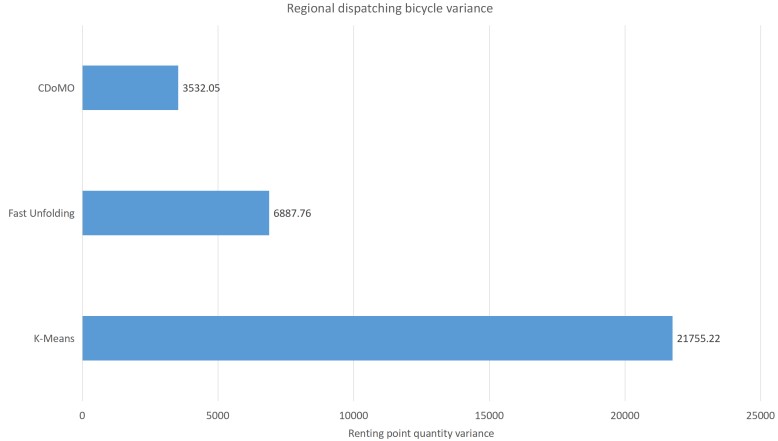

**Figure 18** Distance difference of bicycle scheduling area under three algorithms in the Chicago dataset.

## CONCLUSION

In order to solve the problem of regional division of public bicycles, this paper proposes **CDoMO**. The algorithm fully considers the special law of public bicycle lease/return, and in order to balance the scheduling workload between areas, the regional scheduling workload index is proposed. This problem is identified as a multi-objective optimization problem with two objective functions: minimize the variance between the estimated dispatch distances between each area; minimize the variance between the numbers of sites in each area. The regional scheduling workload can adjust the results of the community discovery algorithm in real time and dynamically. In the end, the results obtained can meet the special rules of public bicycle lease/return, and balance the workload between the areas. The experimental results show that the **CDoMO** can effectively shorten the scheduling

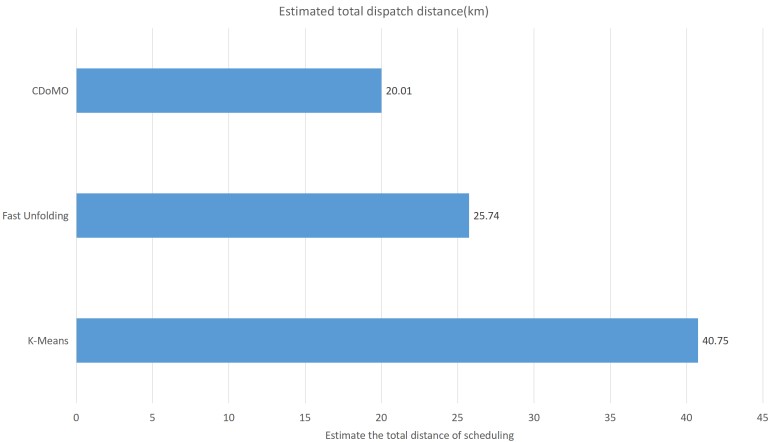

**Figure 19** **Total distance of estimated scheduling under three algorithms in the Chicago dataset.**

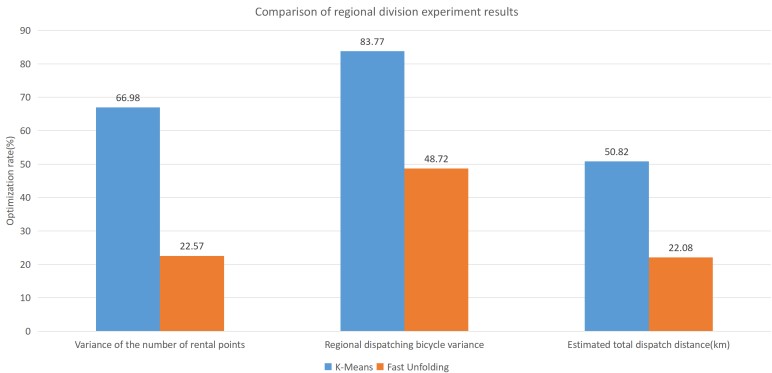

**Figure 20** **Comparison of experimental results of region partition under three algorithms in the Chicago dataset.**

distance of public bicycle system, effectively improve the scheduling efficiency, and make the workload of each scheduling area relatively balanced. The next step is to have a more appropriate solution if you limit the travel time and mileage of the scheduling vehicle.

# ACKNOWLEDGEMENTS

The authors are grateful to the anonymous referee for a careful checking of the details and for helpful comments that improved this paper.

### Funding

Funding for this work was financially supported by the National Natural Science Foundation of China (No. 61602141) and the Key Research and Development Program of Zhejiang Province, China (Grant No.2019C03138). The funders had no role in study design, data collection and analysis, decision to publish, or preparation of the manuscript.

### Grant Disclosures

The following grant information was disclosed by the authors:
National Natural Science Foundation of China: 61602141.
Key Research and Development Program of Zhejiang Province, China: 2019C03138.

### Competing Interests

The authors declare there are no competing interests.

### Author Contributions

- Fei Lin conceived and designed the experiments, analyzed the data, contributed reagents/materials/analysis tools, prepared figures and/or tables, performed the computation work, authored or reviewed drafts of the paper, approved the final draft.
- Yang Yang conceived and designed the experiments, performed the experiments, prepared figures and/or tables, performed the computation work, authored or reviewed drafts of the paper, approved the final draft.
- Shihua Wang analyzed the data, contributed reagents/materials/analysis tools, authored or reviewed drafts of the paper, approved the final draft.
- Yudi Xu performed the experiments, prepared figures and/or tables, performed the computation work, approved the final draft.
- Hong Ma analyzed the data, prepared figures and/or tables, performed the computation work, approved the final draft, research funding.
- Ritai Yu performed the experiments, prepared figures and/or tables, approved the final draft.

### Data Availability

The data is available at GitHub: https://github.com/YannCuz/425.git.
The New York public bicycle data are available in the Citi Bike Trip Histories repository: https://www.citibikenyc.com/system-data and http://datawrapper.dwcdn.net/rreHM/6/.
The Chicago public bicycle data are available in the Divvy Data repository: https://www.divvybikes.com/system-data.

### Supplemental Information

Supplemental information for this article can be found online at http://dx.doi.org/10.7717/peerj-cs.224#supplemental-information.

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
