# Peer review of "Urban public bicycle dispatching optimization method"

_PeerJ Computer Science, doi:10.7717/peerj-cs.224_

## Round 0.1 · original submission · Major Revisions

Thank you for submitting your manuscript. The comments of the referees on your submitted manuscript can be found at the bottom of this message. Based on these comments, I would like to invite you to prepare a revised version of your manuscript.

·

Basic reporting

1) I suggest that the title could be more specific and unnecessary words like ‘research on’ could be removed
2) Line 13, NSGA2 should be written in the full name when it first appears in the article.
3) In introduction, references should be cited to support the authors’ arguments. For example, list reference to support the sentence that the mainstream regional division method is based on the urban administrative area.
4) Authors should summarize the development of relevant research and analyze their disadvantages. Although authors briefly state the limits of existing approaches, the research gap and the main contribution of this article are not clear enough.

Experimental design

no comment

Validity of the findings

1) Experiment results should be analyzed with more details.
2) Could you add a figure to compare routes of rebalancing using different approaches?

Additional comments

1) The English language should be improved.
2) The figure number is not matched in the context.
3) I recommend that figures could be uploaded with higher resolution.

Reviewer 2 ·

Basic reporting

I am pleasantly surprised to see the paper full text accessible on PeerJ website.
Not sure whether to continue with this review, however, would like to do the needful as agreed.

This paper attempts to present a methodology on policy of dispatching the shared bicycle system at transit-stations or terminals. A community discovery algorithm is used to address this problem. The authors claim an improvement in comparison to the results obtained by NSGA algorithm to the tune of 20-50% of estimated scheduled distance.

The paper in general is very relevant and contemporary. Some issues are:

1. Many places authors with first name initials appear in the text,
Line 39, T. Tulabandhula [1]
Line 41, G. Q. Pan [2]

2. Incomplete reference (few samples):
[1] Tulabandhula, T., & Bodas, T. P. (2018). Method and system for dispatching of vehicles in a public transportation network
[13] Doreswamy, & Ghoneim, O. A. (2018). Traffic jams detection and congestion avoidance in smart city using parallel k-means clustering algorithm.
[22] GAO, X., Tian, Y., Sun, B., GAO, X., Tian, Y., & Sun, B. (2018). Multi-objective optimization design of bidirectional flow passage components using rsm and nsga ii: a case study of inlet/outlet diffusion segment in pumped storage power station. Renewable Energy, 115.

3. Unrelated/Irrelevant references
[24] Data was supposedly taken from this paper (see lines 260-2610
"New York Public Bicycle [24] is a people-benefit project launched by the New York City Government. Figure 5.3 displays the spatial distribution of rental sites." The reference mentioned is on crash and helmet related aspects.
[25] Shixiong Jiang, Wei Guan, Zhengbing He, and Liu Yang, “Exploring the Intermodal Relationship between Taxi and
447 Subway in Beijing, China,” Journal of Advanced Transportation, vol. 2018, Article ID 3981845, 14 pages, 2018.
448 https://doi.org/10.1155/2018/3981845.
449 26. Chao Guo, Cheng Gong, Haitao Xu, and

Experimental design

The data as explained in the paper from public sources. This is essentially from New York and Chicago cities.

New York city data (Reference 22) data is not available here

Chicago city data (reference 31)

Chicago public bicycle data [31]. paper is not accessible nor it has a doi to search. Thus I'm unsure of the data availability.

Python Codes are provided with the manuscript, but data is missing.

Validity of the findings

Thus it would be difficult to validate the findings given these limitations. This study amounts to using Technique A is better than rest B, C etc., without gaining any insights into the problem of allocation of bicycles to various points in the city. Also this paper needs to establish why Technique A is better. Sections 4 and 5 needs improvement in terms of this comprehension and being less abstract.

Additional comments

1) The article can be improved by trying to address the issues brought out here in.
2) The pitch (USP) of the paper needs to be highlighted
3) References need to be improved (complete and relevant ones only needed).

Reviewer 3 ·

Basic reporting

The paper proposes an improved Community Discovery algorithm based on Multi-objective Optimization (CDoMO) in order to solve the problem of regional division of public bicycles. The algorithm fully considers the special law of public bicycle lease/return, and in order to balance the scheduling workload between areas, the regional scheduling workload index is proposed. In particular, two objective functions are considered, i.e., minimize the variance between the estimated dispatch distances between each area and minimize the variance between the numbers of sites in each area. Moreover, the proposed algorithm can work in real time and dynamically.

Experimental design

no comment

Validity of the findings

no comment

Additional comments

The paper is well-organised and the exposition is very clear.
Moreover, I really appreciate the figures shown by the authors, even if their resolution needs to be improved.

I have just a little concern about the adopted acronyms. Please, use capital letters for words involved in the acronym.

Reviewer 4 ·

Basic reporting

The article focused on the arimethic of how to get reasnonable dispatching area division, it is a true interesting topic. Unfortunately, the paper has some major weakness like below. 1) The article should include sufficient introduction and background to demonstrate how the work fit into the dispatching research of public bicycle. For example, there are more literature on how to divide dispatching area should be appriciately referenced. Besides, the prediction of bicycle leaseing demand is also a importand reseach point that occpies the whole Section 3.3, 3.4.1 and 3.4.2 in the paper, but there is no any background introduction about it and we know there are tremendous prior literature about it. 2) It is difficult to estimate the main contribution of this paper due to the lack of professional article structure and sufficient introduction. From my view, there lacks deep understanding on the physical issue of how to divide the dispatching area of pyblic bicycle. 3) The algorithem is not descriped properly such as formula (3.11) , (3.12). Why it is the right formua to fit for your research topic? 4) There are many small errors in article writing and orgnanizing, for example, After Section 3.3, the followed should be Section 3.3.1 and 3.3.2 instead of Section 3.4.1 and 3.4.2.

Experimental design

no comment

Validity of the findings

no comment

Additional comments

The article focused on the arimethic of how to get reasnonable dispatching area division, it is a true interesting research. Unfortunately, the paper has some major weakness like below. 1) The article should include sufficient introduction and background to demonstrate how the work fits into the dispatching research of public bicycle. For example, there are more literature on how to divide dispatching area should be appriciately referenced. Besides, the prediction of bicycle leaseing demand is also a importand reseach point that occpies Section 3.3, 3.4.1 and 3.4.2 in the paper, but there is no any background introduction about it and we know there are many prior literature about it. 2) It is difficult to estimate the main contribution of this paper due to the lack of professional article structure and sufficient introduction. From my view, there lacks deep understanding on the physical issue of how to divide the dispatching area of pyblic bicycle. 3) The algorithem is not descriped properly such as formula (3.11) , (3.12). Why it is the right formua to fit for your research topic? 4) There are many small errors in article writing and orgnanizing, for example, After Section 3.3, the followed should be Section 3.3.1 and 3.3.2 instead of Section 3.4.1 and 3.4.2.

---

## Round 0.2 · Minor Revisions

The reviewer 2 still has some concerns. Please improve the paper accordingly.

Reviewer 2 ·

Basic reporting

1) The references issues remain largely unaddressed particularly comments 1, 2 and 3

2) incomplete references 1, 31 etc (in the v1 manuscript)

3) Data availability is not addressed: just the bikesharing programme addresses are given: I do not see link to any data, except the static locations.

4) I would like to know whether the dynamic bike availability at locations are shared by these cities with the researchers.

Experimental design

no comment

Validity of the findings

no comment

Additional comments

No comments

---

## Round 0.3 · accepted · Accept

I found some references' style is not consistent. However, I believe the content has been revised well. Thus, I recommend the ACCEPT decision.